# STAT3-Mediated Promoter-Enhancer Interaction Up-Regulates Inhibitor of DNA Binding 1 (*ID1*) to Promote Colon Cancer Progression

**DOI:** 10.3390/ijms241210041

**Published:** 2023-06-12

**Authors:** Zhike Lin, Ying Liu, Tian Xu, Ting Su, Yingying Yang, Runhua Liang, Songgang Gu, Jie Li, Xuhong Song, Bin Liang, Zhijun Leng, Yangsihan Li, Lele Meng, Yijing Luo, Xiaolan Chang, Dongyang Huang, Lingzhu Xie

**Affiliations:** 1Department of Cell Biology and Genetics, Key Laboratory of Molecular Biology in High Cancer Incidence Coastal Chaoshan Area of Guangdong Higher Education Institutes, Shantou University Medical College, Shantou 515041, China; 19zklin@alumni.stu.edu.cn (Z.L.); 18yliu7@stu.edu.cn (Y.L.); 17txu@alumni.stu.edu.cn (T.X.); tsu1@stu.edu.cn (T.S.); yyy3403@zznu.edu.cn (Y.Y.); 21rhliang@stu.edu.cn (R.L.); 20jli8@stu.edu.cn (J.L.); songxuhong@stu.edu.cn (X.S.); bliang@stu.edu.cn (B.L.); 21zjleng@stu.edu.cn (Z.L.); liysh2015@lzu.edu.cn (Y.L.); 22llmeng@stu.edu.cn (L.M.); 22yjluo@stu.edu.cn (Y.L.); xlchang@stu.edu.cn (X.C.); 2Department of Hepatobiliary Surgery, Cancer Hospital of Shantou University Medical College, Shantou 515041, China; 11sggu@stu.edu.cn; 3Department of General Surgery, First Affiliated Hospital of Shantou University Medical College, Shantou 515041, China; 4Department of Central Laboratory, Cancer Hospital of Shantou University Medical College, Shantou 515041, China

**Keywords:** *ID1*, enhancer, STAT3, colon cancer

## Abstract

Background: High expression of inhibitor of DNA binding 1 (*ID1*) correlates with poor prognosis in colorectal cancer (CRC). Aberrant enhancer activation in regulating *ID1* transcription is limited. Methods: Immunohistochemistry (IHC), quantitative RT-PCR (RT-qPCR) and Western blotting (WB) were used to determine the expression of *ID1*. CRISPR-Cas9 was used to generate *ID1* or enhancer E1 knockout cell lines. Dual-luciferase reporter assay, chromosome conformation capture assay and ChIP-qPCR were used to determine the active enhancers of *ID1*. Cell Counting Kit 8, colony-forming, transwell assays and tumorigenicity in nude mice were used to investigate the biological functions of *ID1* and enhancer E1. Results: Human CRC tissues and cell lines expressed a higher level of *ID1* than normal controls. *ID1* promoted CRC cell proliferation and colony formation. Enhancer E1 actively regulated *ID1* promoter activity. Signal transducer and activator of transcription 3 (STAT3) bound to *ID1* promoter and enhancer E1 to regulate their activity. The inhibitor of STAT3 Stattic attenuated *ID1* promoter and enhancer E1 activity and the expression of *ID1*. Enhancer E1 knockout down-regulated *ID1* expression level and cell proliferation in vitro and in vivo. Conclusions: Enhancer E1 is positively regulated by STAT3 and contributes to the regulation of *ID1* to promote CRC cell progression and might be a potential target for anti-CRC drug studies.

## 1. Introduction

Globally, colorectal cancer (CRC) is the third most common cancer and remains the second leading cause of cancer-related deaths due to its poor survival outcome in advanced stages [1,2]. Therefore, further understanding of the mechanisms of CRC tumorigenesis and progression is required for the development of interventions to prevent and manage CRC progression.

Inhibitors of DNA binding proteins (*ID1*-4) are members of the helix-loop-helix (HLH) transcription factor family (*ID1*-4) [3]. *ID1* proteins are over-expressed in a variety of tumors [4,5,6], in which over-expression of *ID1* was found to be associated with cancer aggressiveness and poor clinical outcomes. *ID1* is generally considered an oncogene and has been reported to mediate the stemness of colorectal cancer cells [4]. By binding to the *ID1* promoter, transcription factors (TFs) STAT3 [7], LEF1 [8], ATF1 [9], and CXXC5 [10] regulate *ID1* transcription. However, the research on other DNA regulatory elements for *ID1* transcriptional regulation in CRC is limited.

Gene transcription is commonly regulated by elements such as enhancers and silencers [11,12]. Enhancers function as integrated TF docking platforms and mediate gene transcription via long-range interactions [13]. TFs dock on enhancers and simultaneously bind to the mediator complex, which binds to the basal transcription machinery at the promoters [14]. In the presence of chromatin loops, TF binding at enhancers activates nearby promoters by diffusing over a small nuclear volume [15]. A combination of enhancer markers (high levels of monomethylated H3 lysine 4 (H3K4me1) and acetylated H3 lysine 27 (H3K27ac)) is used to identify the active states of enhancers [16]. Most active enhancers bi-directionally transcribe RNAs, named enhancer RNAs (eRNAs), whose expression levels positively correlate with mRNA levels of enhancer target genes [17]. Although dysregulation of enhancers usually induces abnormal gene expression that drives the uncontrolled proliferation of human cancers [18], the biological contribution of each enhancer toward gene function remains largely unknown.

By analyzing enhancer histone modification hallmarks, we discovered three putative enhancers located near *ID1*. We hypothesized that these putative enhancers regulate *ID1* expression in colon cancer. To test this hypothesis, we determined the role of the three putative *ID1* enhancers in regulating *ID1* expression and the TFs that regulate *ID1* enhancer activity in colon cancer.

## 2. Results

### 2.1. ID1 Expression Level Is Up-Regulated in CRC

In the ONCOMINE database, the mRNA expression of *ID1* was up-regulated in many cancers, including CRC (Figure 1A). The mRNA level of *ID1* was up-regulated in primary CRC tissues compared with normal controls, with fold-changes of 2.727 in the colon adenoma [19], 3.129 in the rectal adenoma [19] and 2.021 in the colorectal adenocarcinoma [20] (Figure 1B–D). IHC analysis showed that *ID1* protein expression levels were higher in 20 of 23 primary colon cancer tissues compared with adjacent normal tissues (86.96%) (Figure 1E). Elevated expression of *ID1* was also observed in CRC cell lines LoVo, C2BBel, SW620, HT29 and HCT116, as compared with CCD-18Co (Figure 1F,G). Our results are in line with those previously reported [4,21], suggesting that *ID1* expression is increased in CRC tissues and cells.

### 2.2. ID1 Promotes CRC Cell Proliferation and Colony Formation

To explore the biological function of *ID1*, we generated *ID1* knockout HCT116 cells (ID1_KO) using CRISPR-Cas9 (Figure 2A). After genome editing, *ID1* protein was not expressed (Figure 2B), and cell proliferation and colony formation were decreased (Figure 2C,D). In addition, we over-expressed *ID1* in another CRC cell line LoVo by transient transfection. *ID1* was effectively increased at the protein level in the *ID1* over-expression cells (OE_ID1) (Figure 3A). Cell proliferation and colony formation were increased in OE_ID1 cells (Figure 3B,C). These results suggest that *ID1* has a cell proliferation-promoting effect on CRC cells. There was no significant effect of *ID1* expression level on invasion and migration of CRC cells (Figure 2E and Figure 3D).

### 2.3. Identification of ID1-Associated Enhancers in CRC

By searching the ENCODE ChIP-seq data in the UCSC Genome Browser [22], we found three putative enhancer regions located near *ID1* (Figure 4A). All these regions possessed high levels of H3K27ac and H3K4me1 but low levels of tri-methylated H3 lysine 4 (H3K4me3). These three putative enhancers are located 8.2 kb and 5.8 kb upstream and 1.0 kb downstream of the transcriptional start site of *ID1*, and we refer to them as Enhancer 1 (E1), Enhancer 2 (E2) and Enhancer 3 (E3), respectively. GRO-Seq profiles of nascent RNA in HCT116 cells show bidirectional transcription at the three putative enhancer regions (Figure 4A). ChIP-seq data showed that PolII is enriched at the three putative enhancer regions in HCT116 (Figure 4A). We denoted the eRNAs transcribed from the three putative enhancers as E1-, E2- and E3-eRNA. CRC cell lines had elevated eRNA expression levels for all three eRNAs compared to CCD-18Co (Figure 4B). The CCD-18Co and CRC cell lines LoVo and HCT116 were used for subsequent experiments.

We next characterized the histone modifications in the three cell lines by ChIP-qPCR. The three putative enhancer regions harbored H3K27ac and H3K4me1 but a low level of H3K4me3 modifications (Figure 4C–E), which coincides with features of enhancer domains [23]. As shown in Figure 4C, H3K27ac was enriched at all three putative enhancer regions and *ID1* promoter, but the enrichment levels gradually increased in the order of in CCD-18Co, LoVo and HCT116 cells. The enrichment level of H3K27ac is in line with *ID1* and eRNA expression levels among the three cell lines (Figure 1F and Figure 4B). This result implies that the *ID1* promoter and enhancers have higher activities in CRC cell lines, especially in the HCT116 cell line.

A dual luciferase reporter assay was performed to measure *ID1* promoter and enhancer activities. Luciferase activity driven by promoter-luc-E1 increased by 1.63 fold (*p* < 0.05) in LoVo cells and by 3.02 fold (*p*  <  0.05) in HCT116 cells, and luciferase activity of promoter-luc-E3 (E3-1 and E3-2) was about 2-fold (*p*  <  0.05) over promoter-luc plasmid in both LoVo and HCT116 cells (Figure 4F). This result indicates that enhancers E1 and E3 are able to enhance the transcription activity of the *ID1* promoter in CRC cell lines but not the normal colon cell line. Furthermore, enhancer E1 had a stronger enhancer effect on *ID1* promoter activity in HCT116 cells than in LoVo cells.

To identify chromatin loops between putative enhancers and the *ID1* promoter, we performed chromosome conformation capture (3C) on the target region (chr20: 31594349-31612156). Enhancer-promoter interaction frequencies were detected with quantitative PCR (qPCR) using qPCR primers by matching the “Promoter” primer with another fragment primer (among 1-11R). In all three cell lines, E1 (5R) was the only locus to form enhancer-promoter looping with the *ID1* promoter (”Promoter”), and the relative cross-linking efficiency was ranked in increasing order from CCD-18Co, LoVo and HCT116 cells.

Collectively, these data show that E1 is an active enhancer in CRC cell lines HCT116 and LoVo, showing typical histone modifications and enhancer activity, and forming enhancer-promoter looping with the *ID1* promoter, while in CCD-18Co cells, E1 displayed relatively weak enhancer features and had no enhancer activity (based on dual-luciferase reporter assay).

### 2.4. STAT3 Promotes Expression of ID1 and Enhancer Activity in CRC Cells

To explore enhancer function in regulating the expression of *ID1* and identify the TFs that mediate enhancer activity, we first searched the literature for TFs that regulate *ID1* expression, which were then used as candidates to screen for binding to the E1 enhancer region by using Transcription Factor Binding data in the JASPAR 2022 UCSC tracks [24]. Prior reports show that STAT3 up-regulates *ID1* expression by binding to the *ID1* promoter in the glioblastoma [7]. A JASPAR database search revealed that the enhancer E1 region contains a putative STAT3-binding site (Figure 5A). The active form STAT3 is pSTAT3, which forms STAT3 homodimers and then translocates into the nucleus where it initiates the transcription [25,26] and is generally used for measuring STAT3 transcription regulation activity. Stattic is a non-peptidic small molecule that specifically inhibits STAT3. Stattic dose-dependently inhibited STAT3 phosphorylation (Figure 5B right). In HCT116 cells, *ID1* mRNA decreased in parallel with the reduction of pSTAT3 (Figure 5B left). Treatment with 20 µM Stattic for 1 h caused a significant decrease in *ID1* protein levels (Figure 5B). Further, we validated the binding of pSTAT3 at the predicted STAT3 binding sites of E1 and *ID1* promoter by ChIP-qPCR. The ChIP-qPCR results demonstrated a relative enrichment of pSTAT3 at the *ID1* promoter and E1 enhancer region (Figure 5C). When treated with 20 µM Stattic for 1 h, the *ID1* promoter and E1 enhancer regions showed decreased enrichment of pSTAT3, H3K27ac and RNA polymerase II (PolII) (Figure 5D). When treated with Stattic at 10 µM, the luciferase activities of promoter-luc-E1 and promoter-luc plasmid decreased 61% and 29%, respectively; when treated with 20 µM Stattic, the luciferase activities of promoter-luc-E1 and promoter-luc plasmid decreased 81% and 65%, respectively (Figure 5E). This result shows that the luciferase activity of the promoter-luc-E1 plasmid is more sensitive to the decrease of pSTAT3 than the promoter-luc plasmid, suggesting that STAT3 can influence *ID1* promoter activity by regulating E1 enhancer activity. In addition, the 3C assay showed that *ID1* promoter/E1 looping is decreased by Stattic treatment (Figure 5F). These results indicate that, besides direct regulation by binding to the *ID1* promoter, STAT3 regulates the expression of *ID1,* probably by increasing the chromatin looping frequency between the *ID1* promoter and E1 enhancer, as well as the activated state of the *ID1* promoter and E1 enhancer chromatin.

The same validations were performed in another CRC cell line LoVo, which had a lower E1-dependent luciferase activity (Figure 4F). Again, treatment with Stattic inhibited STAT3 phosphorylation and decreased *ID1* mRNA expression (Figure 5G), luciferase activities of promoter-luc-E1 and promoter-luc plasmid (Figure 5I) and *ID1* promoter/E1 looping frequencies (Figure 5J), but the extents of decrease were much less than those in the HCT116 cell line. We speculate that the lower E1 enhancer activity and the lower sensitivity to Stattic are probably related to the low expression level of STAT3 in the LoVo cell line (Appendix A), which probably also led to the low quantity of STAT3 bound at the *ID1* promoter and E1 enhancer region (Figure 5H, had no statistical significance).

The above results support the hypothesis that STAT3 regulates the expression of *ID1* both by directly binding to the *ID1* promoter and by regulating the activity of enhancer E1 to indirectly influence the *ID1* expression level.

### 2.5. E1 Enhancer Knockout Reduces ID1 Expression and Inhibits Proliferation of CRC Cells

The above results show active enhancer E1 can regulate *ID1* expression in CRC. However, the independent roles of the E1 in *ID1* expression are unknown. Therefore we conducted knockout of the E1 DNA fragment (1694 bp) from the genome using CRISPR-Cas9 (Figure 6A,B). E1 knockout (E1_KO) down-regulated the *ID1* expression level (Figure 6C). Since *ID1* promotes CRC cell survival and proliferation (Figure 2 and Figure 3), we characterized the cell proliferation and colony formation of HCT116 E1_KO in vitro and in nude mice. E1 knockout (E1_KO) inhibited proliferation and colony formation of HCT116 cells (Figure 6D,E). Mice were given a subcutaneous injection of HCT116 control cells or E1 knockout HCT116 (E1_KO) cells. At the end of the experiment (day 26), the mice injected with E1_KO cells exhibited smaller tumor sizes and tumor weights than those injected with the HCT116 control cells (Figure 6F–H). No significant differences in body weight were found between the two groups of mice during the course of the experiments (Figure 6I).

## 3. Discussion

Herein, we identified an enhancer, denoted E1, and determined its role in the regulation of *ID1* expression in colon cancer. Our findings are as follows: (i) sctive enhancer E1 located upstream of *ID1* positively regulates *ID1* expression; (ii) STAT3 up-regulates the expression of *ID1* in part due to increasing enhancer activity and promoter-enhancer looping frequencies; (iii) enhancer E1 enhances the proliferation of colon cancer cells by regulating *ID1* expression.

Previous research has shown that enhancers associated with cancer exhibit not only tumor-type specificity but also cell-type specificity [27,28]. Our previous study showed that different CRC cell lines possess different active enhancers in the regulation of the oncogene CYR61 expression [29]. In this study, we found that CRC cell lines HCT116 and LoVo harbor the same enhancer E1, but show different activities, and this status may be partly due to the different levels of STAT3 bound to the *ID1* promoter and E1 enhancer region.

Stattic treatment inhibited STAT3 phosphorylation was inhibited and decreased the enrichment of pSTAT3 at the *ID1* promoter and E1 enhancer region. The decreased enrichment of pSTAT3 was associated with a marked decrease in H3K27ac and PolII enrichment at the same locus. Previous studies revealed that NF-κB can assist pSTAT3 loading at a subset of enhancers, while on the other hand, pSTAT3 can retain NF-κB in the nuclei of cancer cells [30,31,32,33]. Moreover, NF-κB is able to recruit p300/CBP to the chromatin [32] to increase H3K27ac [33]. Thus, we reasoned that perturbation of the binding of STAT3 would perturb other co-activators (such as NF-κB) and then alter the chromatin H3K27 acetylation status. H3K27 acetylation eliminates a positive charge on the surface of nucleosomes and loosens nucleosomes to increase the accessibility of RNA polymerase II or other activators [34]. Therefore, by regulating the H3K27 acetylation and RNA PolII enrichment, STAT3 changes the *ID1* promoter and enhancer activities to increase the *ID1* expression level.

Knockout of enhancer E1 showed that enhancer E1 plays an important role in regulating the expression of *ID1* and the proliferation of CRC cells. Enhancer E1 knockout down-regulated the *ID1* expression level and CRC cell proliferation and colony formation in vivo and in vitro. As previously reported [35,36], one enhancer may regulate several target genes not only through long-range intrachromosomal interactions but also through interchromosomal regulation. We show that enhancer E1 regulates *ID1* expression through chromatin looping. However, we still do not know whether enhancer E1 regulates the expression of other genes. Further research such as 4C-seq [37] will be needed to explore this question.

## 4. Materials and Methods

### 4.1. Patients and Tissues

A total of 23 cases of colonic adenocarcinoma (with corresponding matched normal colonic mucosa) were obtained at the First Affiliated Hospital of Shantou University Medical College (Guangdong, China) from 2015 to 2018. All collected tissues had pathologic diagnoses from two independent pathologists and were histologically classified and staged according to the tumor node metastasis (TNM) system [38]. The clinical characteristics of patients are presented in Appendix A. No patient received radiotherapy or chemotherapy prior to surgery. All subjects gave their informed consent for inclusion before participating in the study. This study was conducted in accordance with the Declaration of Helsinki, and the protocol was approved by the Ethics Committee of Shantou University Medical College (number: SUMC-2015-42).

### 4.2. Immunohistochemistry (IHC)

Antigen unmasking was performed by heat treatment, and sections were incubated with an *ID1* mouse monoclonal primary antibody (1:100; Santa Cruz, TX, USA, sc-133104) overnight at 4 °C. Anti-mouse/rabbit HRP-labeled polymer universal secondary antibody (Maixin, Fuzhou, China, KIT-5010) was applied for 30 min at RT. Slides were then incubated with diaminobenzidine and counterstained with hematoxylin. Image-Pro Plus v.6.0 software (Media Cybernetics, Rockville, MD, USA) was used to assess the area and the integrated optical density (IOD) of the stained region. Mean density = IOD/area. The average mean density for five random fields at 400× magnification was used for *ID1* statistical analysis.

### 4.3. Cell Culture and Treatment

Normal human colon cell line CCD-18Co was cultured in Eagle’s minimum essential medium (ATCC, Manassas, VA, USA). Human colon carcinoma cell line HCT116 was cultured in McCoy’s 5A (modified) medium (Gibco, New York, NY, USA). Both CCD-18Co and HCT116 cell lines were kindly provided by Prof. Yi Guan (Joint Influenza Research Centre (SUMC/HKU), Shantou University Medical College). The LoVo cell line (Cell Bank of the Chinese Academy of Sciences, Shanghai, China) was cultured in an F-12K medium (Sigma-Aldrich, St. Louis, MI, USA). All cell culture media were supplemented with 10% fetal bovine serum, and cells were cultured at 37 °C with 5% CO_2_. HCT116 and LoVo cells were treated with Stattic (Sigma-Aldrich, S7947) at a final concentration of 10 and 20 µM. An *ID1* expression plasmid GV658-ID1 was constructed by GENECHEM (Shanghai, China). Plasmids were transfected at a final concentration of 1.0 μg/mL using FuGENE HD transfection reagent (Promega Wisconsin, Fitchburg, WI, USA, E2312) according to the manufacturer’s protocol.

### 4.4. RNA Extraction, Reverse Transcription Polymerase Chain Reaction (RT-PCR) and Quantitative PCR (qPCR)

Total RNA was isolated using Trizol (Takara, Kyoto, Japan, No.9109) according to the manufacturer’s instructions. RT-PCR was performed using HiScript^®^ II Q RT SuperMix for qPCR with gDNA wiper (Vazyme, Nanjing, China, R223-01). The qPCR was performed using AceQ qPCR SYBR Green Master Mix (Low ROX Premixed) (Vazyme, Q131-02) on a QuantStudio 12K Flex Real-Time PCR System (Thermo Fisher, Waltham, MA, USA) according to the manufacturer’s protocols. Primers are shown in Appendix A.

### 4.5. Western Blot Assay

Proteins were separated on 10% SDS/PAGE and transferred to a PVDF membrane (Sigma-Aldrich, ISEQ00010). Primary antibodies used are shown in Appendix A.

### 4.6. Microarray, ChIP-seq and GRO-seq Data Analyses

The mRNA expression level of *ID1* in different cancers (including different colon cancer datasets) was analyzed using the ONCOMINE gene expression array datasets (www.oncomine.org, accessed on 1 March 2021), downloaded in March 2021, and data were analyzed with a cut-off *p*-value and fold-change defined as 0.0001 and 2.0. PolII ChIP-seq data (ID: 55663) [39] from Cistrome Data (http://dc2.cistrome.org/, accessed on 1 March 2021) was used and visualized in the UCSC Genome Brower. Data for global nuclear run-on sequencing (GRO-seq) of HCT116 was downloaded from the GEO database (GSM1124062) [40]. GRO-seq data analyses was processed and mapped using a pipeline described previously [29] and visualized in the UCSC Genome Brower.

### 4.7. Chromatin Immuno-Precipitation (ChIP)

As we described before [41], the ChIP assay was done using a chromatin immuno-precipitation kit (Sigma-Aldrich, 17-611) according to the manufacturer’s instructions. Immuno-precipitated DNA was characterized using qPCR and normalized with input DNA. Antibodies and the primers used in ChIP analysis are listed in Appendix A.

### 4.8. CRISPR/Cas9-Mediated ID1 and Enhancer E1 Deletion

CRISPR/Cas9 sgRNAs (designed on http://crispor.tefor.net/, accessed on 2 December 2021 [42]) were cloned into the lentiGuide-Puro plasmid vector (Addgene #52963 [43]). The sgRNA primers are listed in Appendix A. HCT116 cells were first infected with LentiCas9-Blast virus (Addgene #52962 [43]) and selected in 8 µg/mL blasticidin (Genomeditech, Shanghai, China, GM-040404) for 1 week and then infected with lentiGuide-Puro virus encoding different sgRNAs. Single cells were sorted into 96-well plates and selected with 2.0 μg/mL puromycin (BBI, Shanghai, China, A610593). To detect sgRNA-directed cleavage in the *ID1* gene and deletion of E1, genomic DNA was extracted and used as a template for PCR (primers are listed in Appendix A). PCR products were gel purified prior to sequencing.

### 4.9. Cell Proliferation Assay

Cell proliferation was determined by viable cell counting or a Cell Counting Kit 8 assay. Briefly, 1 × 10^3^ cells were seeded in each well of 96 well plates. For viable cell counting, manual counting was performed every 24 h by mixing an aliquot of cells 1:1 with 0.4% trypan blue dye (BBI, A601140). For Cell Counting Kit 8 (CCK8, MCE, Jersey City, NY, USA, HY-K0301) assay, 10 µL CCK-8 reagent was added to each well and cultured for another 3.0 h at 37 °C at each time point and then read on a 96-well plate reader at a wavelength of 450 nm.

### 4.10. Colony-Forming Assay

A total of 300 cells were seeded in 6 well plates, incubated for 2 weeks and fixed with 4% paraformaldehyde. Colonies were stained with 0.1% crystal violet (Beyotime, Shanghai, China, C0121) for 30 min and counted. The percent area covered by colonies was measured by Image J 1.46r. The percent area covered by colonies was compared with a control group.

### 4.11. Transwell Assay

Cell migration and invasion assays were performed in a transwell chamber (8 μm pore size, BD Biosciences, Franklin Lakes, NJ, USA). For migration assay, 1.0 × 10^5^ cells were transferred into the upper chamber. For invasion assay, the membrane was coated with Matrigel (BD Biosciences) diluted 1:40, and 2.5 × 10^5^ cells were seeded into the upper chamber. Migrated/invaded cells were fixed with 4% paraformaldehyde and stained with 0.1% crystal violet (Beyotime, C0121) 24 h after seeding.

### 4.12. Dual-Luciferase Reporter Assay

The *ID1* promoter region DNA fragment was amplified and inserted into the pGL3-basic plasmid (Promega, E1751) at XhoI and HindIII restriction sites to produce the promoter-luc vector. Afterwards, the E1 DNA fragment was amplified and inserted into the promoter-luc vector at KpnI and NheI restriction sites to produce the promoter-luc-E1 vector. E2 and E3 (divided into 3 fragments) DNA fragments were amplified and inserted into promoter-luc vector at MluI and XhoI restriction sites to produce promoter-luc-E2, promoter-luc-E3-1, promoter-luc-E3-2 and promoter-luc-E3-3 vectors, respectively. Primers and the amplify loci are shown in Appendix A. Cells were co-transfected with pGL3-derived plasmids and the internal control plasmid pRL-SV40 (Promega, E2231) at a 100:1 ratio in 100 ng total DNA per well. Luciferase activity was determined by dual-luciferase reporter assay system (Promega, E1910).

### 4.13. Chromosome Conformation Capture (3C) Assay

The 3C assay was performed as described previously [37,44], with minor modifications. PstI was used for genomic DNA digestion. The efficiency of PstI digestion is shown in Appendix A. BAC clones used in 3C assays were *ERCC3* (Thermo Fisher, CTD-3251N23) and *ID1* (Thermo Fisher, CTD-2566B14). Primers used in this assay are shown in Appendix A. The R package “Sushi” was used to plot the figures.

### 4.14. Animal Studies

Five- to six-week-old male BALB/c nude mice were purchased from Beijing Vital River Laboratory Animal Technology Co. (Beijing, China). To detect the effects of E1 on the tumorigenicity, 1 × 10^7^ HCT116-Ctrl or HCT116-E1_KO cells were subcutaneously injected into the right upper limb of each mouse (*n* = 8 per group). Mouse body weight was measured once per week. The tumor sizes were measured weekly after injection for the first two weeks and then once every three days. Tumor volume was calculated as follows: Volume = Length × Width^2^/2 [45]. After the experiment, the mice were euthanised by cervical dislocation. All animal experiments were approved by the Animal Experimental Ethics Committee of Shantou University Medical College (approval number: SUMC-2020-214) and carried out at the Animal Experimental Center of Shantou University Medical College.

### 4.15. Statistical Analysis

Statistical analyses were conducted in SPSS Statistics 26.0. All data shown were determined for three independent experiments unless otherwise stated and presented as the mean ± S.D., * *p* < 0.05, ** *p* < 0.01, *** *p* < 0.001.

## 5. Conclusions

Collectively, our observations support a model whereby enhancer E1, which is located upstream of *ID1*, forms a promoter-enhancer chromatin loop with the *ID1* promoter. TFs, such as STAT3, mediate this chromatin looping and affect the activity of both the *ID1* promoter and enhancer E1 to ultimately regulate *ID1* expression (Figure 7). High enhancer activity in a subset of human colon cancers can increase the expression of *ID1* and promote cell proliferation, suggesting that enhancer E1 might be a potential target for anti-CRC drug studies.

## Figures and Tables

**Figure 1 ijms-24-10041-f001:**
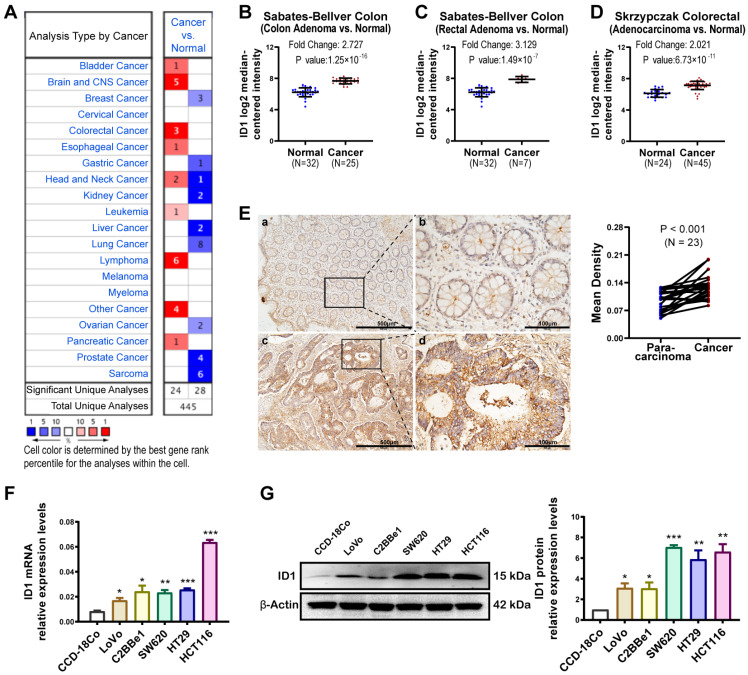
*ID1* expression level in primary colorectal adenocarcinoma patient samples and colon cell lines. (**A**) The mRNA expression level of *ID1* in different cancers based on ONCOMINE gene expression array dataset analysis. (**B**) *ID1* mRNA expression levels in colon adenoma compared with normal controls based on the analysis of the Sabates-Bellver colon microarray data [19]. (**C**) *ID1* mRNA expression levels in rectal adenoma compared with normal controls based on analysis of the Sabates-Bellver colon microarray data [19]. (**D**) *ID1* mRNA expression levels in adenocarcinoma compared with normal controls based on analysis of the Skrzypczak colorectal microarray data [20]. (**E**) Left: immunostaining for *ID1* in paraffin sections of para-carcinoma tissues and cancer tissues from patients with colon adenocarcinoma (magnification, ×100, scale bars  =  500 μm (**a**,**c**); ×400, scale bars  =  100 μm (**b**,**d**)). Right: mean density of *ID1* in IHC in colon para-carcinoma and cancer tissues from 23 cases; significance determined by Student’s *t*-test. (**F**) Expression of *ID1* mRNA and (**G**) protein levels in human colon cell lines; significance determined by one-way ANOVA. Data are shown as mean ± S.D., *n* = 3. * *p* < 0.05, ** *p* < 0.01, *** *p* < 0.001.

**Figure 2 ijms-24-10041-f002:**
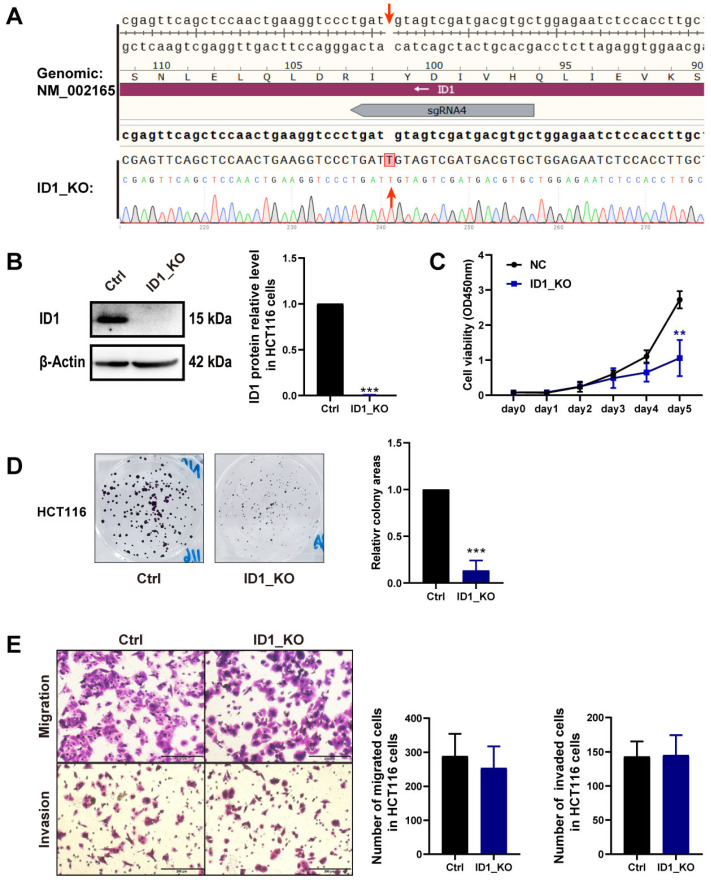
*ID1* knockout decreases HCT116 cell proliferation and colony formation. (**A**) Knockout genomic sequence showing a single base insertion, introduced by genome editing, leading to a frameshift mutation in the *ID1* gene. (**B**) Left: ID1 protein expression was detected using WB after *ID1* knockout. Right: statistic of the relative mean density of *ID1* protein expression level. (**C**) Cell proliferation was quantified with CCK 8 assay after *ID1* knockout. (**D**) Left: colony formation was measured after *ID1* knockout. Right: statistic of the relative colony areas. (**E**) Left: cell migration and invasion were quantified with transwell assays after *ID1* knockout; right: statistic of the number of migrated and invaded cells; Scale bars  =  200 μm. The significance for all data except (**A**) was determined using Student’s *t*-test. Data are shown as mean ± S.D., *n* = 3. ** *p* < 0.01, *** *p* < 0.001.

**Figure 3 ijms-24-10041-f003:**
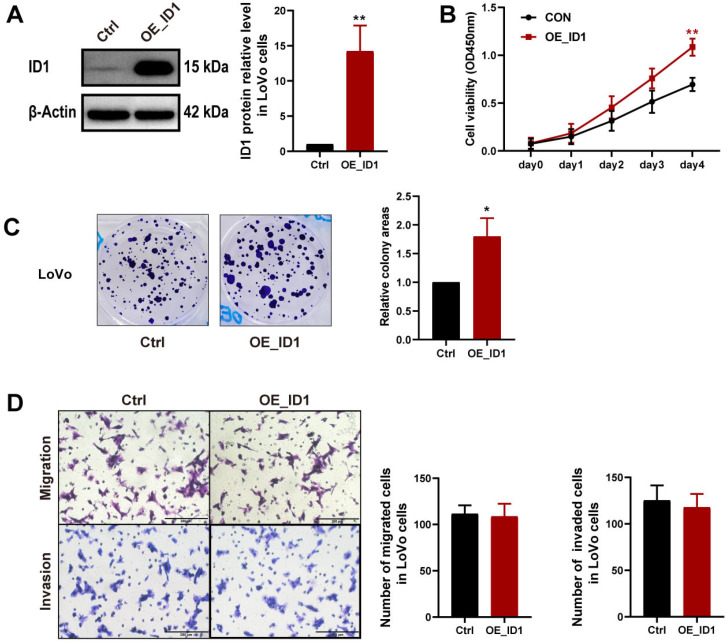
*ID1* over-expression increases LoVo cell proliferation and colony formation. (**A**) Left: ID1 protein expression was quantified using WB after *ID1* over-expression. Right: statistic of the relative mean density of ID1 protein expression level. (**B**) Cell proliferation was detected with CCK-8 assay after *ID1* over-expression. (**C**) Left: colony formation was detected after *ID1* over-expression. Right: statistic of the relative colony areas. (**D**) Left: cell migration and invasion were quantified with transwell assays after *ID1* over-expression; right: statistic of the number of migrated and invaded cells; Scale bars  =  200 μm. The significance of all data was determined using Student’s *t*-test. Data are shown as mean ± S.D., *n* = 3. * *p* < 0.05, ** *p* < 0.01.

**Figure 4 ijms-24-10041-f004:**
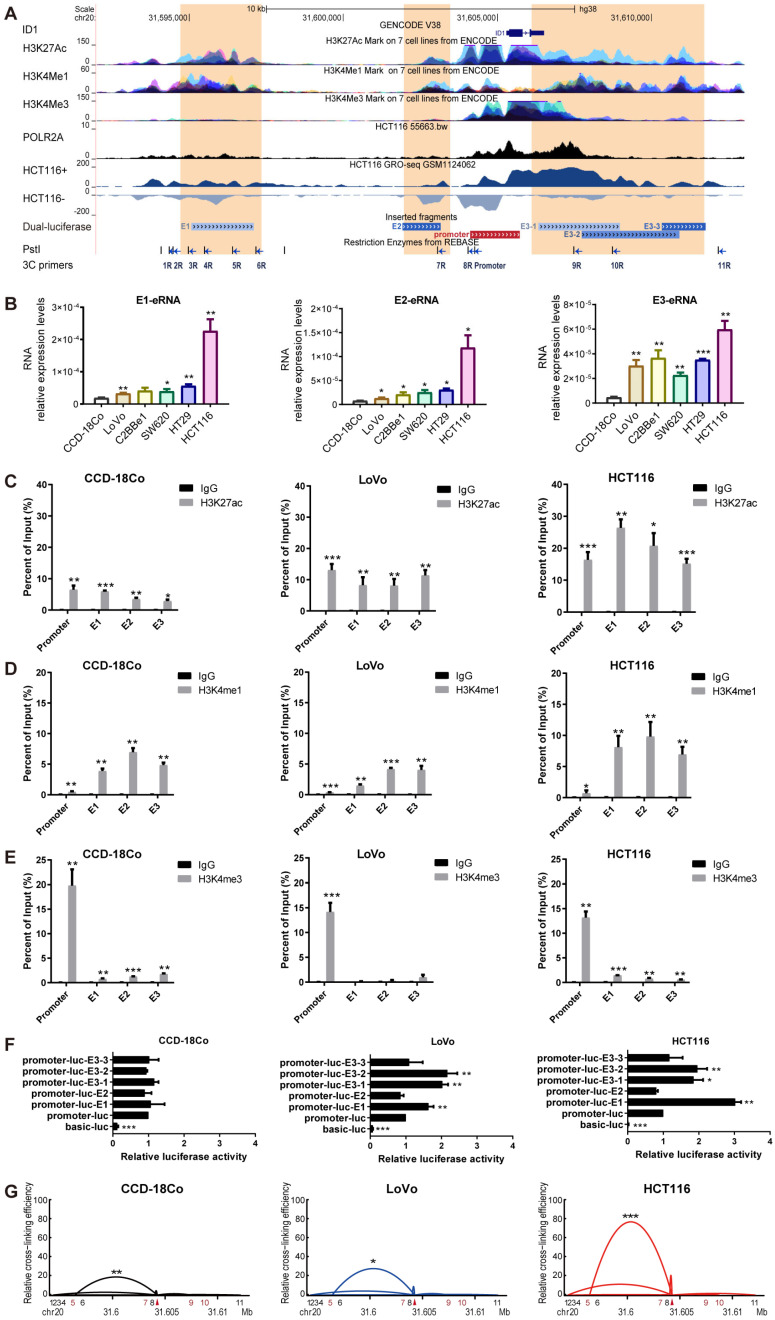
Identification of *ID1* active enhancers. (**A**) From top to bottom: UCSC gene annotation (GRCh38/hg38) of *ID1*; enrichment of H3K27ac, H3K4me1 and H3K4me3 in 7 cell lines from ENCODE; enrichment of PolII in HCT116 cells; GRO-Seq profiles of nascent RNA in HCT116 cells; locations of fragments inserted in the pGL3-basic plasmid; PstI digestion sites and positions of primers used in the 3C assay; the “Promoter” arrow delineates the *ID1* promoter, while arrows 3-5R delineate the E1 locus, arrow 7R delineates the E2 locus and arrows 9-10R delineate the E3 locus. (**B**) Expression levels of eRNAs transcribed from the three putative enhancers in human colon cell lines. (**C**–**E**) Enrichment of H3K27ac, H3K4me1 and H3K4me3 at the *ID1* promoter and the three putative enhancer regions in human colon cell lines, assessed by ChIP-qPCR. (**F**) Relative luciferase reporter activities normalized to the promoter-luc plasmid. (**G**) Relative crosslinking frequencies between the constant region (*ID1* promoter marked with a red triangle) and distal fragments (1R~11R) in the three cell lines, measured by qPCR, normalized to *ERCC3* to calculate the relative cross-linking efficiency. Significance for all data except (**A**) was determined using Student’s *t*-test. Data are shown as mean ± S.D., *n* = 3. * *p* < 0.05, ** *p* < 0.01, *** *p* < 0.001.

**Figure 5 ijms-24-10041-f005:**
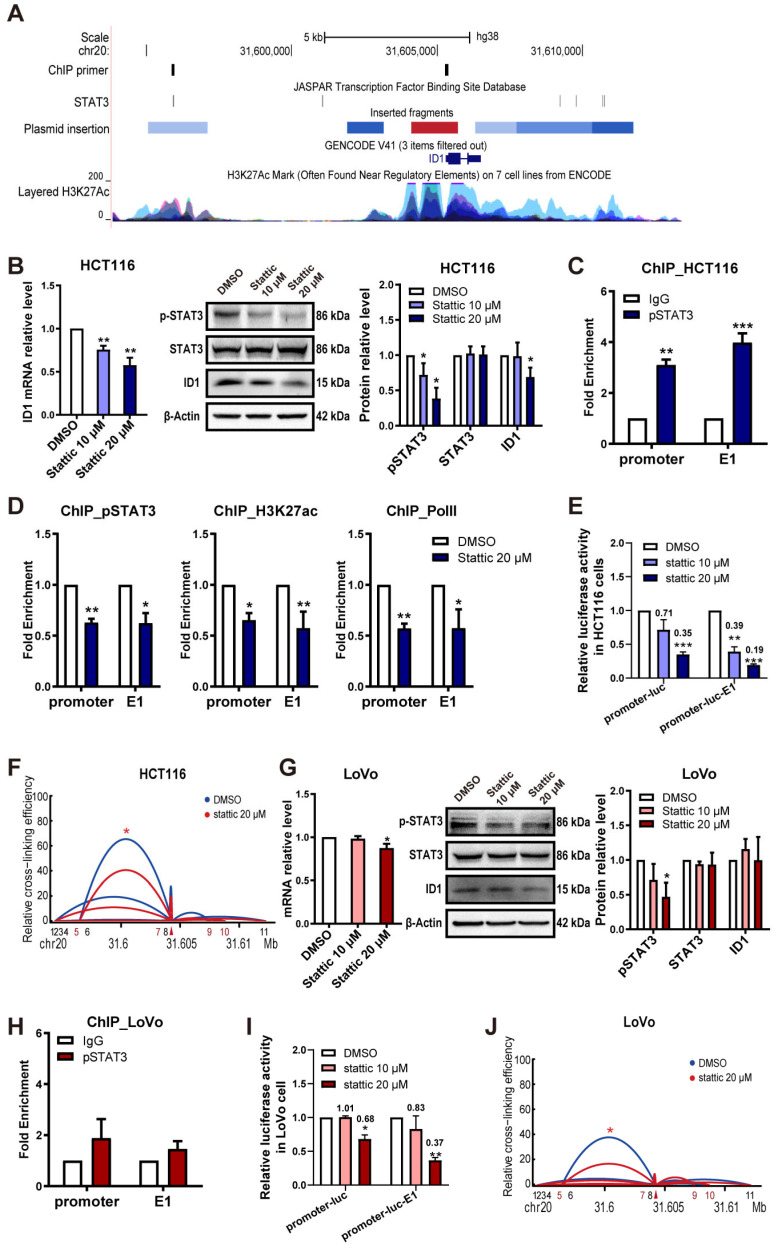
STAT3 inhibitor inhibits expression of *ID1* and enhancer activity in CRC cells. (**A**) From top to bottom: ChIP-qPCR primer positions; binding sites of STAT3 from the Transcription Factor Binding data in JASPAR 2022; locations of fragments inserted in the pGL3-basic plasmid; UCSC gene annotation (GRCh38/hg38) of *ID1*; enrichment of H3K27ac on 7 cell lines from ENCODE; (**B**–**F**) were performed in HCT116 cell line. (**B**) The mRNA and protein expression level of *ID1* after treatment with Stattic for 1 h was assessed by RT-qPCR and WB. (**C**) Enrichment of pSTAT3 at the *ID1* promoter and enhancer E1, assessed by ChIP-qPCR. (**D**) Enrichment of pSTAT3, H3K27ac and PolII at the *ID1* promoter and enhancer E1 after treatment with 20 µM Stattic for 1 h, assessed by ChIP-qPCR. (**E**) Relative luciferase activity of promoter and enhancer E1 after treatment with Stattic for 1 h, as assessed by relative luciferase reporter assay. (**F**) Relative crosslinking frequencies of looping between the *ID1* promoter (marked with a red triangle) and enhancer E1 (fragment 5) after treatment with 20 µM Stattic for 1 h, assessed by 3C; significance determined by paired *t*-test. (**G**–**J**) were performed in the LoVo cell line. (**G**) The mRNA and protein expression level of *ID1* after treatment with Stattic for 1 h was assessed by RT-qPCR and WB. (**H**) Enrichment of pSTAT3 at the *ID1* promoter and enhancer E1, assessed by ChIP-qPCR. (**I**) Relative luciferase activity of the promoter and enhancer E1 after treatment with Stattic for 1 h, as assessed by relative luciferase reporter assay. (**J**) Relative crosslinking frequencies of looping between the *ID1* promoter (marked with a red triangle) and enhancer E1 (fragment 5) after treatment with 20 µM Stattic for 1 h, assessed by 3C; significance determined by paired *t*-test. Significance for all data except (**A**,**F**,**J**) was determined using Student’s *t*-test. Data are shown as mean ± S.D., *n* = 3. * *p* < 0.05, ** *p* < 0.01, *** *p* < 0.001.

**Figure 6 ijms-24-10041-f006:**
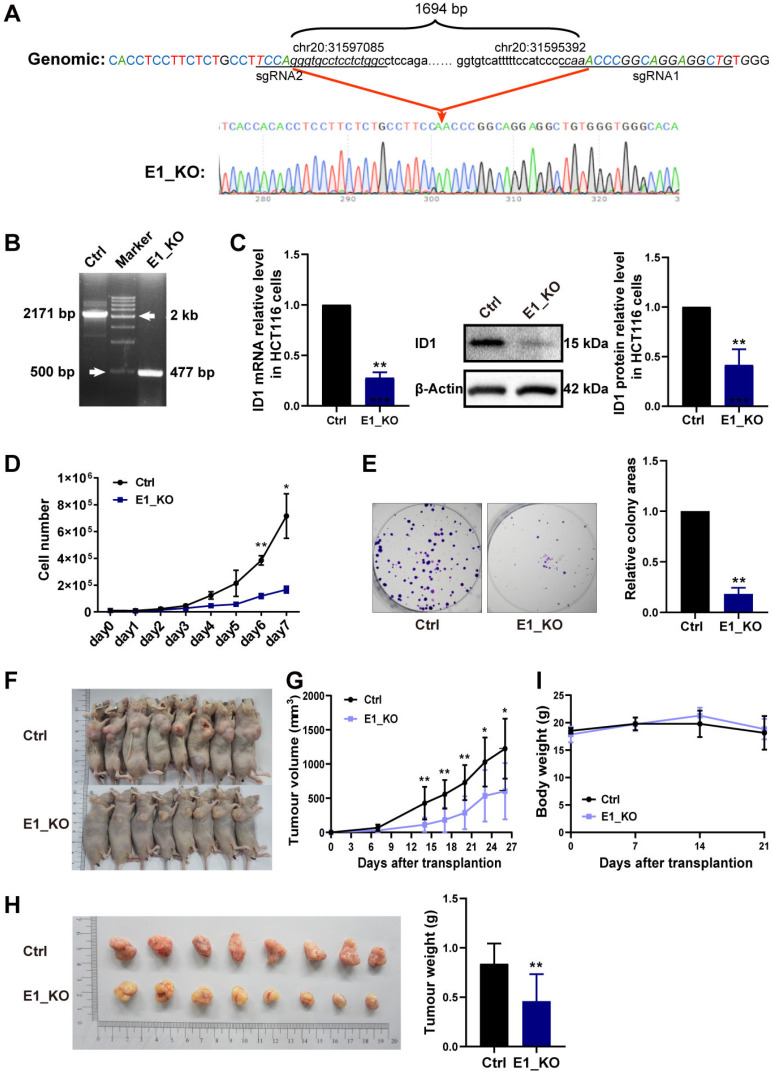
Enhancer E1 knockout reduces expression of *ID1* and inhibits the proliferation of CRC cells. (**A**) The sequence of the E1 enhancer deletion in monoclonal cell line HCT116-E1_KO. (**B**) PCR was used to amplify the E1 region (including the knockout region) in genomic DNA. (**C**) The mRNA and protein expression level of ID1 after E1 knockout, assessed by RT-qPCR and WB. (**D**) Cell proliferation was measured by viable cell counting in control and E1_KO cells. (**E**) Left: colony formation was detected in control and E1_KO cells. Right: statistic of the relative colony areas. (**F**) Images of nude mice with xenograft tumors in each group. (**G**) Tumor volume curve of each group was measured on the indicated days after injection. (**H**) Left: excised tumors imaged from nude mice in each group; right: tumor weights of each group were determined. (**I**) Mouse body weight was measured once per week in each group. Significance for all data except (**A**,**B**,**F**) was determined using Student’s *t*-test. Data are shown as mean ± S.D., *n* = 3. * *p* < 0.05, ** *p* < 0.01.

**Figure 7 ijms-24-10041-f007:**
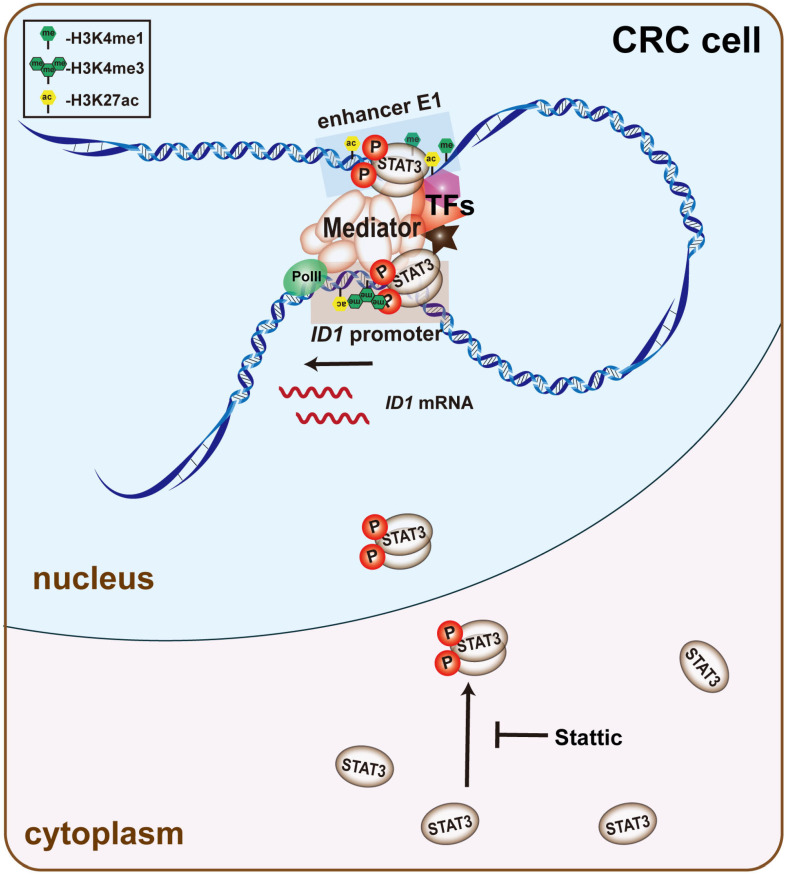
Schematic representation of enhancer E1 regulates ID1 expression. Enhancer E1, which is located upstream of *ID1*, forms a promoter-enhancer chromatin loop with the *ID1* promoter. TFs, such as STAT3, mediate this chromatin looping and affect the activity of both the *ID1* promoter and enhancer E1 to ultimately regulate *ID1* expression.

## Data Availability

Not applicable.

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
