# Peer review of "STAT3-Mediated Promoter-Enhancer Interaction Up-Regulates Inhibitor of DNA Binding 1 (ID1) to Promote Colon Cancer Progression"

_ijms, 2023, doi:10.3390/ijms241210041_

Round 1

Reviewer 1 Report

The presented manuscript is concerning a novel mechanisms by which STAT3-mediated promoter-enhancer interaction can up-regulate the inhibitor of DNA binding 1 to promote progression of colon cancer. The authors revealed that enhancer E1 is positively regulated by STAT3 and contributes to the regulation of ID1 to promote CRC cell progression, and might be a potential target for anti-CRC drug studies.

Paper is written in a thoughtful and understandable way. It briefly summarizes the aim of the study and is divided into individual sections in which the authors accurately explain the carried out research. In terms of content, the information was presented fairly and accurately. What is more, clear and extremely carefully made figures deserve special mention.

The work is an extremely valuable report, however a few minor changes could enrich the manuscript:

1. Particular attention was paid to references. Only 17 from 48 publications are from last 5 years.

2. Were there some exclusion criteria used for patients with CRC?

3. I encourage to give the information about received cell lines in acknowledgements.

4. There are many errors throughout the whole text: typos, different font sizes and types, double spaces, etc.

 The presented manuscript is the valuable and interesting original article and I recommend this paper for publication after minor revisions.

Reviewer 2 Report

The authors of the original article investigated enhancer activation in the transcriptional regulation of ID1 by gene and protein expression methods, CRISPR-Cas9 k.o. cell line generation, luciferase, and chromosome fixation assays, in vitro cell experiments, and mouse models using multiple cell lines. It was found that ID1 expression is higher in CRC and contributes to tumor cell proliferation and colony formation. They identified an E1 enhancer that regulates the activity of the ID1 promoter. STAT3 promotes the function of this E1 enhancer. This suggests that the E1 enhancer plays a role in CRC cell proliferation and progression, and is therefore potentially a molecule that may be worth targeting for therapeutic purposes. 

The experiment is logically structured, well-designed, and the results are presented clearly. The results are understandable, well-supported, and validated. Their conclusions are clear, logical, and moderate. 
Minor comment: 
In subsection 2.1, why ID1 mRNA expression was highest in the adenoma stage. Could this have to do with the known and reported fact that increased methylation of gene promoters also occurs in the adenoma stage of CRC? 
